# Learning to Detect Objects with a 1 Megapixel Event Camera

**Etienne Perot**
PROPHESEE, Paris
eperot@prophesee.ai

**Pierre de Tournemire**
PROPHESEE, Paris
pdetournemire@prophesee.ai

**Davide Nitti**
PROPHESEE, Paris
dnitti@prophesee.ai

**Jonathan Masci**
NNAISENSE, Lugano
jonathan@nnaisense.com

**Amos Sironi**
PROPHESEE, Paris
asironi@prophesee.ai

## Abstract

Event cameras encode visual information with high temporal precision, low data-rate, and high-dynamic range. Thanks to these characteristics, event cameras are particularly suited for scenarios with high motion, challenging lighting conditions and requiring low latency. However, due to the novelty of the field, the performance of event-based systems on many vision tasks is still lower compared to conventional frame-based solutions. The main reasons for this performance gap are: the lower spatial resolution of event sensors, compared to frame cameras; the lack of large-scale training datasets; the absence of well established deep learning architectures for event-based processing. In this paper, we address all these problems in the context of an event-based object detection task. First, we publicly release the first high-resolution large-scale dataset for object detection. The dataset contains more than 14 hours recordings of a 1 megapixel event camera, in automotive scenarios, together with 25M bounding boxes of cars, pedestrians, and two-wheelers, labeled at high frequency. Second, we introduce a novel recurrent architecture for event-based detection and a temporal consistency loss for better-behaved training. The ability to compactly represent the sequence of events into the internal memory of the model is essential to achieve high accuracy. Our model outperforms by a large margin feed-forward event-based architectures. Moreover, our method does not require any reconstruction of intensity images from events, showing that training directly from raw events is possible, more efficient, and more accurate than passing through an intermediate intensity image. Experiments on the dataset introduced in this work, for which events and gray level images are available, show performance on par with that of highly tuned and studied frame-based detectors.

## 1 Introduction

Event cameras [1, 2, 3, 4] promise a paradigm shift in computer vision by representing visual information in a fundamentally different way. Rather than encoding dynamic visual scenes with a sequence of still images, acquired at a fixed frame rate, event cameras generate data in the form of a sparse and asynchronous events stream. Each event is represented by a tuple $(x, y, p, t)$ corresponding to an illuminance change by a fixed relative amount, at pixel location $(x, y)$ and time $t$, with the polarity $p \in \{0, 1\}$ indicating whether the illuminance was increasing or decreasing. Fig. 1 shows examples of data from an event camera in a driving scenario.

Since the camera does not rely on a global clock, but each pixel independently emits an event as soon as it detects an illuminance change, the events stream has a very high temporal resolution, typically

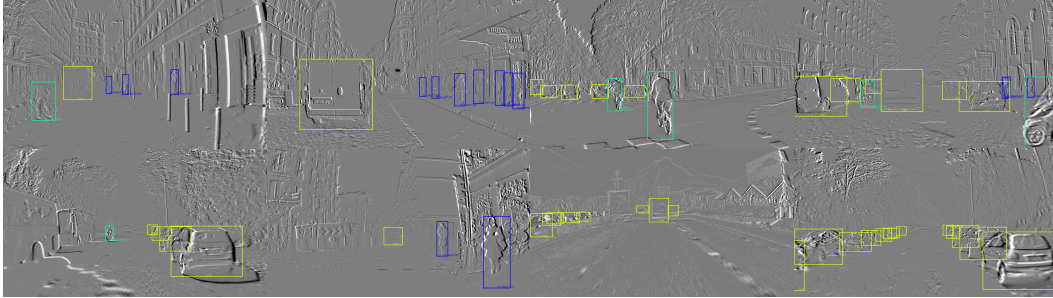

Figure 1: Results of our event-camera detector on examples of the released 1Mpx Automotive Detection Dataset. Our method is able to accurately detect objects for a large variety of appearances, scenarios, and speeds. This makes it the first reliable event-based system on a large-scale vision task. Detected cars, pedestrians and two-wheelers are shown in yellow, blue and cyan boxes respectively. All figures in this work are best seen in electronic form.

of the order of microseconds [1]. Moreover, due to a logarithmic pixel response characteristic, event cameras have a large dynamic range (often exceeding $120dB$) [4]. Thanks to these properties, event cameras are well suited for applications in which standard frame cameras are affected by motion blur, pixel saturation, and high latency.

Despite the remarkable properties of event cameras, we are still at the dawn of event-based vision and their adoption in real systems is currently limited. This implies scarce availability of algorithms, datasets, and tools to manipulate and process events. Additionally, most of the available datasets have limited spatial resolution or they are not labeled, reducing the range of possible applications [5, 6].

To overcome these limitations, several works have focused on the reconstruction of gray-level information from an event stream [7, 8, 9, 10]. This approach is appealing since the reconstructed images can be fed to standard computer vision pipelines, leveraging more than 40 years of computer vision research. In particular, it was shown [10] that all information required to reconstruct high-quality images is present in the event data. However, passing through an intermediate intensity image comes at the price of adding considerable computational cost. In this work, we show how to build an accurate event-based vision pipeline without the need of gray-level supervision.

We target the problem of object detection in automotive scenarios, which is characterized by important objects dynamics and extreme lighting conditions. We make the following contributions to the field: First, we acquire and release the first large scale dataset for event-based object detection, with a high resolution (1280×720) event camera [4]. We also define a fully automated labeling protocol, enabling fast and cheap dataset generation for event cameras. The dataset we release contains more than 14 hours of driving recording, acquired in a large variety of scenarios. We also provide more than 25 million bounding boxes of cars, pedestrians and two-wheelers, labeled at 60Hz.

Our second contribution is the introduction of a novel architecture for event-based object detection together with a new temporal consistency loss. Recurrent layers are the core building block of our architecture, they introduce a fundamental memory mechanism needed to reach high accuracy with event data. At the same time, the temporal consistency loss helps to obtain more precise localization over time. Fig. 1 shows some detections returned by our method on the released dataset. We show that directly predicting the object locations is more efficient and more accurate than applying a detector on the gray-level images reconstructed with a state-of-the-art method [10]. In particular, since we do not impose any bias coming from intensity image supervision, we let the system learn the relevant features for the given task, which do not necessarily correspond to gray-level values.

Finally, we run extensive experiments on ours and another dataset for which gray-level images are also available, showing comparable accuracy to standard frame-based detectors and improved state-of-the-art results for event-based detection. To the best of our knowledge, this is the first work showing an event-based system with on par performance to a frame-based one on a large vision task.

## 2   Related Work

Several machine learning architectures have been proposed for event cameras  [11, 12, 13]. Some of these methods, such as Spiking Neural Networks [14, 15, 16, 17], exploit the sparsity of the data and

can be applied event by event, to preserve the temporal resolution of the events stream [15, 18, 19, 20]. However, efficiently applying these methods to inputs with large event rate remains difficult. For these reasons, their efficacy has mainly been demonstrated on low-resolution classification tasks.

Alternative approaches map the events stream to a dense representation [21, 22, 23, 10]. Once this representation is computed, it can be used as input to standard architectures. Even if these methods lose some of the event's temporal resolution, they gain in terms of accuracy and scalability [24, 22].

Recently, the authors of [10] showed how to use a recurrent UNet [25] to reconstruct high-quality gray-level images from event data. The results obtained with this method show the richness of the information contained in the events. However, reconstructing a gray-level image before applying a detection algorithm adds a further computational step, which is less efficient and less accurate than directly using the events, as we will show in our experiments.

Very few other works have focused directly on the task of event-based object detection. In [18], the authors propose a sparse convolutional network inspired by the YOLO network [26]. While in [27], temporally pooled binary images from the event camera are fed to a faster-RCNN [28]. However, these methods have only been tested on simple sequences, with a few moving objects on a static background. As we will see, feed-forward architectures are less accurate in more general scenarios.

The lack of works on event-based object detection is also related to the scarce availability of large benchmarked datasets. Despite the increasing effort of the community [29, 30, 31, 32], very few datasets provide ground-truth for object detection. The authors of [33] provide a pedestrian detection dataset. However, it is composed of only 12 sequences of 30 seconds. Simulation [34, 35] is an alternative way to obtain large datasets. Unfortunately, existing simulators use too simplified hardware models to accurately reproduce all the characteristics of event cameras. Recently [5] released an automotive dataset for detection. However, it is acquired with a low-resolution QVGA event camera and it contains low frequency labels ($\leq$ 4Hz). We believe instead that high-spatial resolution and high-labeling frequency are crucial to properly evaluate an automotive detection pipeline.

## 3  Event-based Object Detection

In this section, we first formalize the problem of object detection with an event camera, then we introduce our method and the architecture used in our experiments.

### 3.1  Problem Formulation

Let $\mathbf{E} = \{e_i = (x_i, y_i, p_i, t_i)\}_{i \in \mathbb{N}}$ be an input sequence of events, with $x_i \in [0, M]$ and $y_i \in [0, N]$ the spatial coordinates of the event, $p_i \in \{0, 1\}$ the event's polarity and $t_i \in [0, \infty)$ its timestamp. We characterize objects by a set of bounding boxes $\mathbf{B} = \{b_j^* = (x_j, y_j, w_j, h_j, l_j, t_j)\}_{j \in \mathbb{N}}$, where, $(x_j, y_j)$ are the coordinates of the top left corner of the bounding box, $w_j, h_j$ its width and height, $l_j \in \{0, \dots, L\}$ the label object class, and $t_j$ the time at which the object is present in the scene.

A general event-based detector is given by a function $\mathcal{D}$, mapping $\mathbf{E}$ to $\mathbf{B} = \mathcal{D}(\mathbf{E})$. Since we want our system to work in real time, we will assume that the output of a detector at time $t$ will only depend on the past, i.e. on events generated before $t$: $\mathcal{D}(\mathbf{E}) = \{D(\{e_i\}_{t_i < t})\}_{t >= 0}$, where $D(\{e_i\}_{t_i < t})$ outputs bounding boxes at time $t$. In this work, we want to learn $D$.

Applying the detector $D$ at every incoming event is too expensive and often not required by the final applications, since the apparent motion of objects in the scene is typically much slower than the pixels response time. For this reason, we only apply the detector at fixed time intervals of size $\Delta t$:

$$\mathcal{D}(\mathbf{E}) \approx \{D(\{e_i\}_{t_i < t_k})\}_{k \in \mathbb{N}}, \tag{1}$$

with $t_k = k \Delta t^1$. However, a function $D$ working an all past events $\{e_i\}_{t_i < t_k}$ for every $k$, would be computationally intractable, since the number of input events would indefinitely increase over time.

A solution would be to consider at each step $k$, only the events in the interval $[t_{k-1}, t_k)$, as it is done for example in [24, 22] for other event-based tasks. However, as we will see in Sec. 5, this approach leads to poor results for object detection. This is mainly due to two reasons: first, it is hard to choose a single $\Delta t$ (or a fixed number of events) working for objects having very different speeds and sizes,

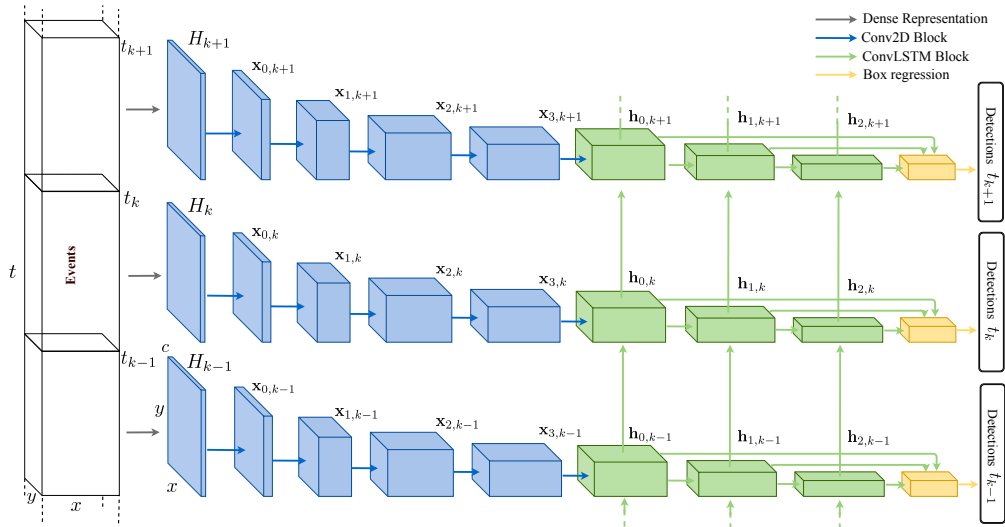

Figure 2: Overview of the proposed architecture. Input events are used to build a tensor map $H_k$ at every time step $t_k$. Feed-forward convolutional layers extract low-level features from $H_k$. Then, ConvLSTM layers extract high-level spatio-temporal patterns. Finally, multiscale features from the recurrent layers are passed to the output layers, to predict bounding box locations and class. Thanks to the memory of the ConvLSTM layers, temporal information is accumulated and preserved over time, allowing robust detections even when objects stop generating events in input.

such as cars and pedestrians. Secondly, since events contain only relative change information, an event-based object detector must keep a memory of the past. In fact, when the apparent motion of an object is zero, it does not generate events anymore. Tracking objects using hard-coded rules is generally not accurate for edge cases such as reflections, moving shadows or object deformations.

For these reasons, we decide to learn a memory mechanism end-to-end, directly from the input events. To use past events information while keeping computational cost tractable, we choose $D$ such that

$$\mathcal{D}(\mathbf{E}) \approx \{D(\{e_i\}_{t_i \in [t_{k-1}, t_k)}, \mathbf{h}_{k-1})\}_{k \in \mathbb{N}}, \qquad (2)$$

where $\mathbf{h}_{k-1}$ is an internal state of our model encoding past information at time $t_{k-1}$. For each $k$, we define $\mathbf{h}_k$ by a recursive formula $\mathbf{h}_k = F(\{e_i\}_{t_i \in [t_{k-1}, t_k)}, \mathbf{h}_{k-1})$, with $\mathbf{h}_0 = \mathbf{0}$. In the next sections, we describe the recurrent neural network architecture we propose to learn $D$ and $F$.

## 3.2 Method

In this section, we describe the recurrent architecture we use to learn the detector $D$. In order to apply our model, we first preprocess the events to build a dense representation. More precisely, given input events $\{e_i\}_{t_i \in [t_{k-1}, t_k)}$ we compute a tensor map $H_k \in \mathbb{R}^{C \times M \times N}$, with C the number of channels. We denote $H_k = H$ in the following. Our method is not constrained to a particular $H$ (cfr. Sec. 5.1).

To extract relevant features from the spatial component of the events, $H$ is fed as input to a convolutional neural network [36, 37]. In particular, we use Squeeze-and-Excitation layers [38], as they performed better in our experiments. In addition, we want our architecture to contain a memory state to accumulate meaningful features over time and to remember the presence of objects, even when they stop generating events. For this, we use ConvLSTM layers [39], which have been successfully used to extract spatio-temporal information from data [40, 41].

Our model first uses $K_f$ feed-forward convolutional layers to extract high-level semantic features that are then fed to the remaining $K_r$ ConvLSTM layers (cfr. Fig. 2). This is to reduce the computational complexity and memory footprint of the method due to recurrent layers operating on large feature maps, and more importantly to avoid the recurrent layers to model the dynamics of low-level features that is not necessary for the given task. We denote this first part of the network *feature extractor*.

The output of the feature extractor is fed to a bounding box *regression head*. In this work, we use Single Shot Detectors (SSD) [37], since they are a good compromise between accuracy and

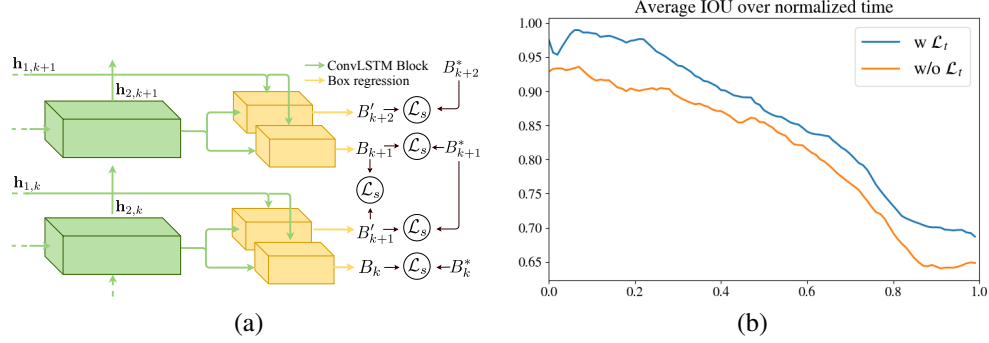

Figure 3: (a) Detail of the box regression heads. In order to regularize temporally our network, we introduce a secondary regression head, predicting, at time $t_k$, the boxes $B'_{k+1}$ for time $t_{k+1}$. We impose predictions corresponding to the same time step to be consistent. (b) IoU between ground truth tracks and predicted boxes over time. The consistency loss helps obtaining more precise boxes.

computational time. However, our feature extractor could be used in combination with other detector families, such as two-stage detectors. Since we want to extract objects for a large range of scales, we feed features at different resolutions to the regression head. In practice, we use the feature map from each of the recurrent layers. A schematic representation of our architecture is provided in Fig. 2.

As typically done for object detection, to train the parameters of our network, we optimize a loss function composed of a regression term $\mathcal{L}_r$ for the box coordinates and a classification term $\mathcal{L}_c$ for the class. We use smooth $l1$ loss [37] $\mathcal{L}_s$ for regression and the softmax focal loss [42] for classification. More precisely, for a set of J ground-truth bounding boxes at time $t_k$, we encode their coordinates in a tensor $B^*$ of size $(\text{J} \cdot \text{R}, 4)$, as done in [37], where R is the number of default boxes of the regression head matching a ground-truth box. Let $(B, p)$ be the output of the regression head, with $B$ the tensor encoding the prediction for the above R default boxes and $p$ the class probability distribution for all default boxes. Then, the regression and classification terms of the loss are:

$$\mathcal{L}_r = \mathcal{L}_s(B, B^*), \quad \mathcal{L}_c = -(1 - p_l)^\gamma \log p_l, \tag{3}$$

where $p_l$ is the probability of the correct class $l$. We set the constant $\gamma$ to 2 and also adapt the unbalanced biases for softmax logits in the spirit of [42].

### 3.2.1 Dual Regression Head and Temporal Consistency Loss

To have temporally consistent detections, we would like the internal states of the recurrent layers to learn high-level features that are stable over long periods of time. Even if ConvLSTM can, to some extent, learn slow-changing representations, we further improve detections consistency by introducing an auxiliary loss and an additional regression head trained to predict bounding boxes one time-step into the future. The idea is inspired by unsupervised learning methods such as word2vec [43] and CPC [44], which constraint the latent representation to preserve some domain structure. In our case, given that features are shared for both heads, we argue that this has the additional effect of inducing representations that account for object motion, something which is currently used as regularization, but would require further analysis that goes beyond the scope of the current work.

Given the input tensor $H_k$ computed in the time interval $[t_{k-1}, t_k)$, the two regression heads will output bounding boxes $B_k$ and $B'_{k+1}$, trying to match ground truth $B^*_k$ and $B^*_{k+1}$ respectively. This dual regression mechanism is shown in Fig. 3(a).

To train the two regression heads, we add to our loss an auxiliary regression term between $B'_{k+1}$ and $B^*_{k+1}$. This term, when applied at every time step $k$, indirectly constraints the output $B'_k$'s of the second head to be close to the predictions $B_k$'s of the first head at the next time step, cfr. Fig. 3(a). However, since the two heads are independent, they could converge to different solutions. Therefore, we further regularize training by adding another loss term explicitly imposing $B'_k$ to be close to $B_k$. In summary, the auxiliary loss is:

$$\mathcal{L}_t = \mathcal{L}_s(B'_{k+1}, B^*_{k+1}) + \mathcal{L}_s(B'_k, B_k). \tag{4}$$

Then, the final loss we use during training is given by $\mathcal{L} = \mathcal{L}_c + \mathcal{L}_r + \mathcal{L}_t$. We minimize it during training using truncated backpropagation through time [45].

# 4   The 1 Megapixel Automotive Detection Dataset

In this section, we describe an automated protocol to generate datasets for event cameras. We apply this protocol to generate the detection dataset used in our experiments. However, our approach can be easily adapted to other computer vision tasks, such as face detection and 3D pose estimation.

**Setup and Fully Automated Labeling Protocol**   The key component to obtaining automated labels is to do recordings with an event camera and a standard RGB camera side by side. Labels are first extracted from the RGB camera and then transferred to the event camera pixel coordinates by using a geometric transformation. In our work, we used the 1 megapixel event camera of [4] and a GoPro Hero6. The two cameras were fixed on a rigid mount side by side, as close as possible to minimize parallax errors. For both cameras, we used a large field of view:  110 degrees for the event camera and  120 degrees for the RGB camera.  The video stream of the RGB camera is recorded at 4 megapixels and 60fps. Once data are acquired from the setup, we perform the following label transfer: 1. Synchronize the time of the event and frame cameras; 2. Extract bounding boxes from the frame camera images; 3. Map the bounding box coordinates from the frame camera to the event camera. The bounding boxes from the RGB video stream are obtained using a commercial automotive detector, outperforming freely available ones. The software returns labels corresponding to pedestrians, two-wheelers, and cars. The time synchronization can be done using a physical connection between the cameras. However, since this is not always possible, we also propose in the supplementary material an algorithmic way of synchronizing them. Once the 2 signals are synchronized temporally, we need to find a geometric transformation mapping pixels from the RGB camera to the event camera. Since the distance between the two cameras is small, the spatial registration can be approximated by a homography. Both time synchronization and homography estimation can introduce some noise in the labels. Nonetheless, we observed time synchronization errors smaller than the discretization step $\Delta t$ we use, and that the homography assumption is good enough for our case, since objects encountered in automotive scenarios are relatively far compared to the cameras baseline. We discuss more in depth failure cases of the labeling protocol in Sec. 5.3. More details can be also be found in the supplementary material.

**Recordings and Dataset Statistics**   Once the labeling protocol is defined, we can easily collect and label a large amount of data. To this end, we mounted the event and frame cameras behind the windshield of a car. We asked a driver to drive in a variety of scenarios, including city, highway, countryside, small villages, and suburbs. The data collection was conducted over several months, with a large variety of lighting and weather conditions during daytime. At the end of the recording campaign, a total of 14.65 hours was obtained. We split them in 11.19 hours for training, 2.21 hours for validation, and 2.25 hours for testing. The total number of bounding boxes is 25M. More statistics can be found in the supplementary material, together with examples from the dataset. To the best of our knowledge, the presented event-based dataset is the largest in terms of labels and classes. Moreover, it is the only available high-resolution detection dataset for event cameras[2].

# 5   Experiments

In this section, we first evaluate the importance of the main components of our method in an ablation study. Then, we compare it against state-of-the-art detectors. We consider the COCO metrics [46] and we report COCO mAP, as is it widely used for evaluating detection algorithms. Even if this metric is designed for frame-based data, we explain in the supplementary material how we extend it to event data. Since labeling was done with a 4 Mpx camera, but the input events have lower resolution, in all our experiments, we filter boxes with diagonal smaller than 60 pixels. All networks are trained for 20 epochs using ADAM [47] and learning rate 0.0002 with exponential decay of 0.98 every epoch. We then select the best model on the validation set and apply it to the test set to report the final mAP. [3]

## 5.1   Ablation Study

As explained in Sec. 3.2, our network can take different representations as input. Here we compare the commonly used Histograms of events [48, 22], Time Surfaces [49], and Event Volumes [23]. The results are given in Tab. 1. We see that Event Volume performs the best. Time Surface is 2% points less accurate than Event Volume, but more accurate than simple Histograms. We notice that we could also learn the input representation $H$ together with the network. For example by combining it

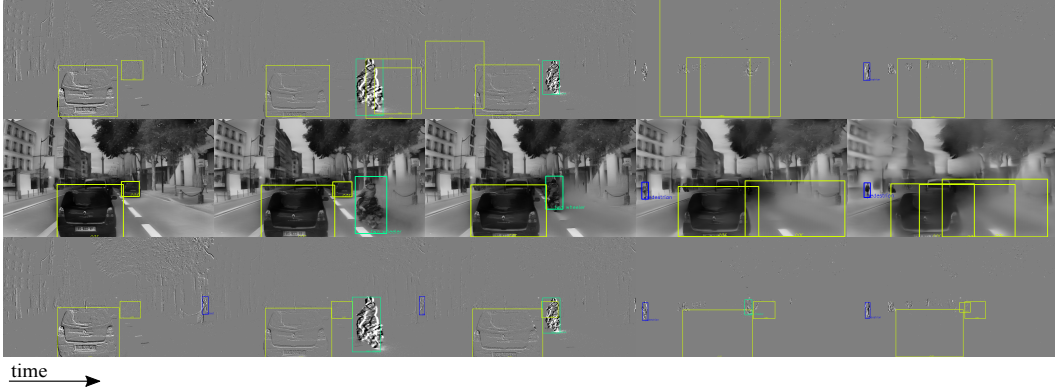

time ⟶

Figure 4: Detections on a 1 Mpx Dataset sequence. From top to bottom: **Events-RetinaNet**, **E2Vid-RetinaNet** (with the input reconstructed images), and our method **RED**. Thanks to the memory representation learned by our network, **RED** can detect objects even when they stop generating events, as for example the stopped car on the right, even when occluded by the motorbike.

with [50, 51, 52]. However, for efficiency reasons, we decided to use a predefined representation and introduce the memory mechanism in the deeper layers of the network, rather than at the pixel level.

In a second set of experiments, we show the importance of the internal memory of our network. To do so, we train while constraining the internal state of the recurrent layers to zero. As we can see in Tab. 1, the performance drops by 12%, showing that the memory of the network is fundamental to reach good accuracy. Finally, we show the advantage of using the loss $\mathcal{L}_t$ of Sec. 3.2.1. When training with this term, mAP increases by 2%, and COCO mAP$_{75}$ increases by 4%, Tab. 1. This shows the advantage of using $\mathcal{L}_t$, especially with regard to box precision. In order to better understand the impact of the loss $\mathcal{L}_t$ on the results, we compute the Intersection over Union (IoU) between 1000 ground truth tracks from the validation set and the predicted boxes. We normalize the duration of the tracks to obtain the average IoU. As shown in Fig. 3(b), with $\mathcal{L}_t$, IoU is higher for all tracks duration.

Table 1: Ablation study on the 1Mpx Dataset. **Left**: mAP for different input representations (without consistency loss). **Right**: mAP and mAP$_{75}$ without some components of our method, with Event Volume as input. "w/o memory" means forcing the internal state $\mathbf{h}_k$ to be zero, for all recurrent layers.

| Histogram | Time Surface | Event Volume | | w/o memory | w/o $\mathcal{L}_t$ loss | memory + $\mathcal{L}_t$ loss |
|---|---|---|---|---|---|---|
| 0.37 | 0.39 | 0.41 | | 0.29 \| 0.26 | 0.41 \| 0.40 | 0.43 \| 0.44 |

## 5.2 Comparison with the State-of-the-art

We now compare our method with the state-of-the-art on the 1 Mpx Detection Dataset and the Gen1 Detection Dataset [5], which is another automotive dataset acquired with a QVGA event camera [53].

We denote our approach **RED** for Recurrent Event-camera Detector. For these experiments, we consider Event Volumes of 50ms. Since there are not many available algorithms for event-based detection, we use as a baseline a feed-forward architecture applied to the same input representation as ours, thus emulating the approach of [33] and [27]. We considered several architectures, leading to similar results. We report here those of RetinaNet [42] with ResNet50 [54] backbone and a feature pyramid scheme, since it gave the best results. We refer to this approach as **Events-RetinaNet**. Then, we consider the method of [10], which is currently the best method to reconstruct graylevel images from events and uses a recurrent Unet. For this, we use code and network publicly released by the authors. Then, we train the RetinaNet detector on these images. We refer to this approach as **E2Vid-RetinaNet**. For all methods, before passing the input to the first convolutional layer of the detector, input height and width are downsampled by a factor 2. For the Gen1 Detection Dataset, we report also results available from the literature [13].

Finally, since the 1 Mpx Dataset was recorded together with a RGB camera, we can train a frame-based detector on these images. Since events do not contain color information, we first convert

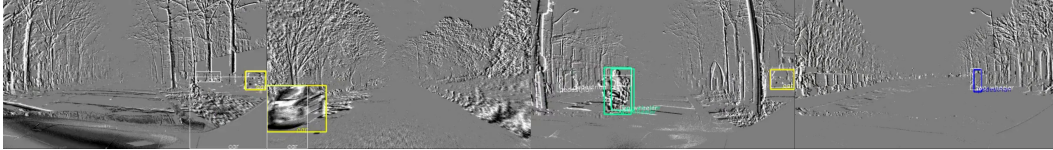

Figure 5: Labeling and detector failure cases. White boxes correspond to labels, colored boxes to detections. From left to right: Labels outlier, labels misalignment, double detection, label swap.

the RGB images to grayscale. Moreover, to have the same level of noise in the labels due to the automated labeling, we map frame camera pixels to the same resolution and FOV as the event camera. In this way, we can have an estimation of how a grayscale detector would perform on our dataset. Similarly, since the Gen1 Dataset was acquired using an event camera providing gray levels, we could run the RetinaNet detector on them. We refer to this approach as **Gray-RetinaNet**.

The results we obtain are given in Tab. 2. We also report the number of parameters of the networks and the methods runtime, including both events preprocessing and detector inference, on a i7 CPU at 2.70GHz and a GTX980 GPU. Qualitative results are provided in Fig. 4 and in the supplementary material. From Fig. 4, we see in particular that our model continues detecting the car even when it does not generate events. While **Events-RetinaNet** becomes unstable and **E2Vid-RetinaNet** begins oversmoothing the image, and thus loses the detection. As we can see from Tab. 2, our method outperforms all the other event-based ones by a large margin. On the 1Mpx dataset the images reconstructed by [10] are of good quality and therefore **E2Vid-RetinaNet** is the second best method, even if 18% points behind ours. Instead, on the Gen1 Dataset, the model of [10] does not generalize well and images are of lower quality. Therefore, on this dataset, **Events-RetinaNet** scores better. Our method reaches the same mAP as **Gray-RetinaNet** on the 1Mpx Dataset, making it the first event-camera detector with comparable precision to that of commonly used frame-camera detectors. If we also consider color, the RetinaNet mAP increases to 0.56, confirming that color information is usefull to increase the accuracy. Our method could benefit from color if acquired by the sensor, such as in [55]. On the Gen1 Dataset our method performs slightly worse, this is due to the higher level of noise of the QVGA sensor and also because the labels lower frequency makes training a recurrent model more difficult. Finally, we observe that our method has less parameters than the others, it runs realtime and, on the 1Mpx Dataset, it is 21x faster than **E2Vid-RetinaNet**, which reconstructs intensity images.

Table 2: Evaluation on the two automotive detection datasets.

|  | 1Mpx Detection Dataset | | | Gen1 Detection Dataset | |
|---|---|---|---|---|---|
|  | mAP | runtime (ms) | params (M) | mAP | runtime (ms) |
| **MatrixLSTM** [50] | - | - | - | 0.31* | - |
| **SparseConv** [13] | - | - | - | 0.15 | - |
| **Events-RetinaNet** | 0.18 | 44.05 | 32.8 | 0.34 | 18.29 |
| **E2Vid-RetinaNet** | 0.25 | 840.66 | 43.5 | 0.27 | 263.28 |
| **RED** (ours) | **0.43** | **39.33** | **24.1** | 0.40 | **16.70** |
| **Gray-RetinaNet** | **0.43** | 41.43 | 32.8 | **0.44** | 17.35 |

\* Provided by the authors, using a pretrained YOLOv3.

### 5.3 Failure Cases

In this section, we discuss some of the failure cases of our network and of the automated labeling protocol of Sec. 4. In Fig. 5, we show the results of our detector together with the ground truth on some example sequences. In the ground truth we observe two types of errors: geometric errors and semantic errors. Geometric errors, such as misalignment between objects and bounding boxes are due to an imprecise temporal and spatial registration between the event and the frame cameras. Semantic errors, such as label swaps or erroneous boxes, are due to wrong detections of the frame based software used for labeling. Our detector can correct some of the geometric errors, if the errors in the box position are uniformly distributed around the objects. Semantic labels are harder to correct, and an outlier robust loss might be beneficial during training. In addition, we observe that our detector can produce double detections and is less accurate on small objects.

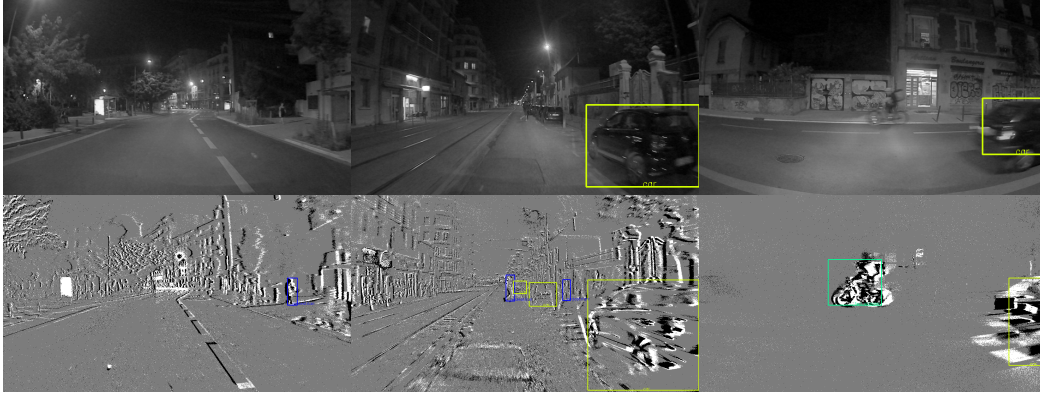

Figure 6: **Top**: **Gray-Retinanet** applied to night recordings of a HDR automotive camera. **Bottom**: Our detector **RED** applied to recordings of a 1 Mpx event camera of the same scene. The detectors were trained on day light data. **Gray-Retinanet** does not generalize well on night images. In contrast, **RED** generalizes on night sequences because event data is invariant to absolute illuminance levels.

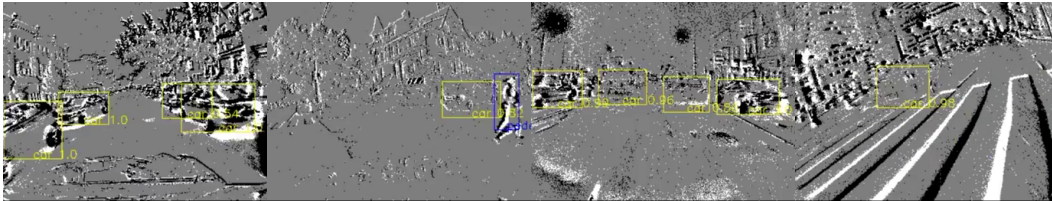

Figure 7: **RED** detector trained on ATIS data and applied to DAVIS sequences. Even if our model was trained on a different camera, it generalizes to other sensors, points of view and light conditions.

## 5.4 Generalization to Night Recordings and Other Event Cameras

We now study the generalization capabilities of our detector. First, we focus on applying our detector, trained with daylight data only, on night recordings. Since event cameras are invariant to absolute illuminance levels, an event-based detector should generalize better than a frame-based one. To test this, we apply **RED** and **Gray-RetinaNet** detectors on new recorded night sequences, captured using the event camera of Sec. 4 and a HDR automotive camera. We stress the fact that these networks have been trained exclusively on daylight data. Since the frame-based labeling software of Sec. 4 is not accurate enough for night data, we report qualitative results in Fig. 6 and in the appendix. It can be observed that the accuracy of **Gray-RetinaNet** drops considerably. This is due to the very different lighting and the higher level of motion blur inherently present in night sequences. On the contrary, our method performs well also in these conditions.

In a second experiment, we test the generalization capability of our network when using different camera types as input. Since there is no available dataset with object detection labels, we report qualitative results on the MVSEC dataset [32], which is an automotive dataset acquired with a DAVIS-346 camera. For this purpose we use the model trained on the Gen1 Dataset, since it was acquired with an ATIS camera with similar resolution as the DAVIS. From Fig. 7, we see that even if the model was trained on a different camera, it generalizes well also on the DAVIS sequences.

## 6 Conclusion

We presented a high-resolution event-based detection dataset and a real-time recurrent neural network architecture which can detect objects from event cameras with the same accuracy as mainstream gray-level detectors. We showed it is possible to consistently detect objects over time without the need for an intermediate gray-level image reconstruction. However, our method still needs to pass through a dense event representation. This means that our method does not take advantage of the input data sparsity. In the future, we plan to exploit the sparsity of the events to further reduce computational cost and latency. This could be done for example by adapting our method to run on neuromorphic hardware [56, 57].

## Broader Impact

The integration of an event-based object detection pipeline in real-world applications could positively impact several aspects of existing systems. First, the camera's high temporal resolution would allow faster reaction time and be more robust in situations where standard cameras suffer from motion blur or high latency. Secondly, they could also improve performance in HDR or low light scenes. Both these aspects are essential to increase the safety of driving assistance solutions or autonomous vehicles [58]. Similarly, these characteristics could be useful in applications where there is an interaction between humans and robots (e.g., in a production line or in a warehouse). Finally, the adoption of similar pipelines in other contexts, like the Internet of Things, could reduce the power consumption and the data storage of existing systems [59].

Although, as demonstrated in [10], the possibility to reconstruct intensity images from events stream could create privacy issues, the proposed method allows better privacy management. Encoding events in not human-readable structures and not requiring to have an image-like representation prevents the easy use of the recorded data for purposes different than those defined by the original algorithm, and it limits the possibility to identify people, vehicles or places.

Further advances in event-based processing and neuromorphic architectures might also open the future to a new class of extremely low-power and low-latency artificial intelligence systems [60]. In a world where power-hungry deep learning techniques are becoming a commodity, and at the same time, environmental concerns are increasingly pressuring our way of life, neuromorphic systems could be an essential component of a sustainable society [61].

Concerning possible negative outcomes, since our method relies on training data, it will leverage the bias and the limitations contained in it. Similarly, since it relies on deep learning architectures, it might be deceived by adversarial attacks. To mitigate these consequences, several methods have been recently proposed to de-bias deep learning models and make them more robust to adversarial examples [62, 63, 64, 65]. A failure of the system might cause dangerous incidents and have severe consequences on people and facilities [66]. Similarly, its integration in a fully autonomous vehicle, poses the ethical question of replacing the human morale in the so called Trolley Problem [67]. Moreover, autonomous vehicles may impact the careers of millions of people [68].

Finally, we think it is essential to be aware that the event-based perception and similar detection systems could be exploited to harm people and threaten human rights. For example, developing modified versions of this algorithm for mass surveillance [69] or military applications [70, 71].

## Acknowledgments and Disclosure of Funding

We would like to thank Davide Migliore for the organization and the support inside the company during the whole duration of paper preparation. We would also like to thank Cécile Pignon for acquiring and reviewing most of the sequences of 1 Megapixel dataset. Finally, thank to Matteo Matteucci and Marco Cannici from Politecnico di Milano for fruitful discussions and for providing the MatrixLSTM results of Table 2.

This publication and the related work was performed in the scope of the ES3CAP research project, under the Bpifrance Invest for the Future Program (*Programme d'Investissements d'Avenir* — PIA). This work was also funded in part by the EU NEOTERIC H2020-ICT-2019-2 project 871330.

## Footnotes

[1]Our method can also be applied on a fixed number of events. For clarity, we only describe the fixed $\Delta t$ case.

[2]Dataset available at: prophesee.ai/category/dataset/

[3]Evaluation code at github.com/prophesee-ai/prophesee-automotive-dataset-toolbox

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
