[Supplementary Material · Learning_to_detect_object_with_1_Mpx_EB_cam_suppl_camera_ready.pdf]

# Supplementary Material for the Paper:
# Learning to Detect Objects with a 1 Megapixel Event Camera

**Etienne Perot**
PROPHESEE, Paris
eperot@prophesee.ai

**Pierre de Tournemire**
PROPHESEE, Paris
pdetournemire@prophesee.ai

**Davide Nitti**
PROPHESEE, Paris
dnitti@prophesee.ai

**Jonathan Masci**
NNAISENSE, Lugano
jonathan@nnaisense.com

**Amos Sironi**
PROPHESEE, Paris
asironi@prophesee.ai

In this appendix, we report additional details which did not fit in our submission due to space limitations. In Sec. 1, we provide more statistics about the released 1 Mpx Automotive Detection Dataset and details about the automated labeling protocol, introduced in Sec. 4 of the main submission. Then, in Sec. 2, we explain how to adapt the COCO metric for an event-based object detection task. In Sec. 3 and Sec. 4, we formally define the losses and the input representations used in our experiments. Finally, in Sec. 5 we provide the architecture used in our experiments,

Attached to this appendix, we also provide a video with the results of our method on sample sequences from the two detection datasets used in our experiments and a comparison with the frame-based detector on night recordings.

## 1 The 1 Megapixel Automotive Detection Dataset

As described in our submission, we build our dataset thanks to a fully automated labeling protocol for event cameras. In this section, we provide more statistics about our dataset and more details about the automated labeling protocol.

**Dataset Statistics**   The 1Mpx Automotive Detection Dataset is composed of 14.65 hours or recordings of a 1280x720 event camera [1]. Data are recorded in different scenarios, including city, with different level of traffic, highway, countryside, small villages and suburbs. The data collection campaign was conducted over several months, with a large variety of lighting and weather conditions during daytime. Recordings are labeled with the automated protocol described in the main submission, yielding 25 million bounding boxes of cars, pedestrians and two-wheelers (i.e. bikes and motorbikes). We split the recordings in 11.19 hours for train, 2.21 hours for validation and 2.25 hours for test. Moreover, to ease training and evaluation each recording is split in 60 seconds chunks, leaving at least one minute between chunks belonging to the different splits. Precise statistics are given in Tab. 1. Sample images form the dataset are given in Fig 1.

Table 1: Number of labels per class on the 1 Mpx Automotive Detection Dataset.

|            | **Car**    | **Pedestrian** | **Two-wheeler** |
|------------|------------|------------|-------------|
| Train      | 11,567,763 | 6,022,634  | 802,707     |
| Validation | 2,339,095  | 919,596    | 138,934     |
| Test       | 2,396,966  | 1,487,702  | 171,428     |
| Total      | 16,303,824 | 8,429,932  | 1,113,069   |

Figure 1: **Left**: RGB frames used for labelling, with overlaid bounding boxes returned by the automotive labeling software. **Right**: sample snapshots from the 1Mpx Detection Dataset, with transferred bounding boxes. The bounding boxes are transferred from the RGB camera using the automoated labeling protocol described in Sec. 1.

**Time Synchronization and Spatial Registration**    The labeling protocol consists in transferring labels from RGB camera data to event camera data, recorded side by side. In the following, we describe an algorithmic way to synchronize the two cameras and to obtain the homography mapping RGB pixels coordinates to event camera pixel coordinates.

Time synchronization is obtained by using zero-normalized cross-correlation (ZNCC) applied on one-dimensional statistics extracted from the two signals. The 1D signals extracted from the event-based camera are the sum and the standard deviation of the number of events per pixel in a time slice of $1/60s$. The 1D signals extracted from the RGB camera are the sum and standard deviation of the intensity value of the absolute difference of consecutive gray-level frames. For each pair of signals (e.g., sum of events and sum of intensity of frame difference) the ZNCC is computed and the highest value is the estimate of temporal difference between the event-based camera and the RGB camera. All estimates are finally averaged using the median.

To estimate the homography, we extract feature points from the two data streams. For the RGB stream, we simply extract FAST [2] points. For the event camera, feature points are obtained by extracting Harris points from histogram of events in a time slice of length $1/60s$. Once feature points are obtained, we can fit an homography using standard methods, such as RANSAC [3].

An alternative solution to estimate the homography consists in minimizing the squared difference between two images (pixel by pixel) where one image is the histogram of events and the other image is the difference of gray level frames. Such loss is then minimized using gradient descent.

## 2 Evaluation Methodology

In this section, we describe how we adapt the COCO metric protocol to work with event data.

We first notice that for the datasets we consider, ground-truth labels are given by a set of bounding boxes at fixed frequency. As a consequence, if the detections returned by an algorithm have the same frequency as the ground truth, accuracy computation is equivalent to evaluating a frame-based detection algorithm.

For the more general case in which detections have a different rate than the ground truth, or when detections are returned in an asynchronous fashion, evaluation can be reduced to the previous case. This is achieved by restricting evaluation to only timestamps for which both detections and ground-truth information are available. If necessary one can add a small time tolerance on the detections timestamps.

For both datasets we consider, the ground-truth is annotated starting from gray-level images. For this reason, there might be some ground-truth bounding boxes at the beginning of a sequence for which no events have been generated yet (such as a static object in front of the sensor when the recording car is stopped). Since it would be impossible for an event-based algorithm to predict the presence of an object in this particular condition, we decide to ignore such boxes during training and evaluation in the first 0.5 seconds of each sequence. More precisely, no loss is computed during training for $0.5s$ and the validation is skipped for the same amount of time.

Similarly, since the frame-based camera of Sec. 1, used for annotation has larger resolution, very far objects are not clearly distinguishable in the event camera. For this reason, we ignore bounding boxes with diagonal size smaller then 60 pixels or smaller then 20 pixel in width or height, during training and evaluation.

## 3 Losses

In this section, we formally define the losses used in the paper.

### 3.1 Softmax Focal Loss

For the classification term $\mathcal{L}_c$ of our loss, we consider the Focal Loss [4]. In the original paper, the authors of [4] used a one-versus-all setting with sigmoid. Instead, in our experiments, we obtain better results using a softmax setting.

The focal loss allows to concentrate on hard examples more efficiently in highly unbalanced cases such as semantic segmentation or object detection. To do so, the individual terms of the cross-entropy loss are weighted with $(1 - p_l)^\gamma$ where $p_l$ is the probability of the correct class. It has the effect that a big error term will count exponentially more, compensating for the rare classes. One problem arises at the beginning of training, where most probabilities are random. Therefore, most of negative examples will still overwhelm the loss for a lot of iterations. As a solution, the authors of [4] provide an efficient biased initialization to increase probability of classifying everything as negative. In their one-versus-all setting, each logit predicts a class versus background (the probability is computed with sigmoid), meaning there is no competition among logits for a single object. They initialize biases of all logits $s_l$ with $-\log \frac{p_x}{1-p_x}$, where $p_x$ is set to 0.01. In this way, after applying the sigmoid, the initial probability to predict the background class is around $1 - p_x = 0.99$ (it would be exactly 0.99 if the weights were zero).

We adapt the same principle to initialize the probability in the case of a softmax classification. We set the logit biases of non-background classes $s_x = 0$ and the logit bias for background $s_\emptyset = \log(C \frac{p_\emptyset}{1-p_\emptyset})$ with $p_\emptyset = 0.99$ and $C$ the number of object classes (excluding background). Applying the softmax leads to a probability for background class close to $p_\emptyset = 0.99$. To understand this let us assume that the probability of a (non-background) class $p_x$ is equal for each class $p_x = \frac{1-p_\emptyset}{C}$ and from the softmax formula we know that $p_x = \frac{e^{s_x}}{e^{s_\emptyset}+Ce^{s_x}}$. Developing the following gives rise to difference of logits: $s_\emptyset - s_x = \log(C \frac{p_\emptyset}{1-p_\emptyset})$. Therefore we use this formula for bias of background and set the biases of other classes $s_x$ to zero.

### 3.2 Smooth l1 loss

For the regression loss $\mathcal{L}_r$ and the auxiliary loss $\mathcal{L}_t$, we consider the smooth $l1$ loss [5], denoted as $\mathcal{L}_s$. The loss $\mathcal{L}_s(B, B^*)$ applied to a tensor $B$ and ground truth $B^*$ is given by the following:

$$\mathcal{L}_s(B, B^*) = \frac{1}{N} \sum_j \mathcal{L}_s(B_j, B_j^*) \tag{1}$$

$$\mathcal{L}_s(B_j, B_j^*) = \begin{cases} |B_j - B_j^*| - \frac{\beta}{2} & \text{if } |B_j - B_j^*| \geq \beta \\ \frac{1}{2\beta}(B_j - B_j^*)^2 & \text{otherwise} \end{cases} \tag{2}$$

where $B_j$ and $B_j^*$ are the elements of $B$ and $B^*$ respectively. In the experiments, we set $\beta = 0.11$.

## 4 Input Representation

In this section, we formally define the input tensor representations we consider in the Ablation Study of the main submission, namely Histograms of events [6, 7], Time Surfaces [8], and Event Volumes [9]. In the following we assume we are given an input sequence of events $\{e_i = (x_i, y_i, p_i, t_i)\}_{i=1}^I$ in a time interval of size $\Delta t$.

### 4.1 Histogram

A Histogram of events is simply given by the sum of events per pixels. We also sum the events independently per polarity, by using one channel per polarity. We clamp at maximum value $m$ of events per pixels and then we divide by this maximum value. This leads to a tensor input of size $(2, M, N)$, where a generic element is the following:

$$H_{p,x,y} = \min \left( 1, \sum_{i, x_i=x, y_i=y, p_i=p} 1/m \right). \tag{3}$$

In our experiments, we use $m = 20$.

### 4.2 Timesurface

A Timesurface is a 2D snapshot of the latest timestamps of events for a given receptive field. It can be seen as a proxy to the normal optical flow [10]. If we accumulate event's absolute timestamps on this array, we obtain a buffer containing all latest timestamps per pixel since the beginning of the record. To give less weight to old events, we apply an exponential decay with parameter $\tau_j$ to the timestamps. Thus, assuming for simplicity $t_0 = 0$, the timesurface $T$ is given by:

$$T_{p,j,x,y} = \exp \left( \frac{ts_{p,x,y} - \max_{x,y}(ts_{p,x,y})}{\tau_j} \right), \tag{4}$$

$$ts_{p,x,y} = \max_{i, x_i=x, y_i=y, p_i=p} t_i. \tag{5}$$

In order to account for slow and fast motion, we consider two decays $\tau_j$ of 10 and 100 ms. This way we can see both motion gradients in very recent and also moderately far in the past without need of any rolling buffer. The formulation being differentiable, we could also learn the set of decay constants $\tau_j$, but we let this for future work. The timesurface input shape used in the experiments is $(4, M, N)$ where the first dimension refers to the 2 decays for each polarity.

### 4.3 Event Volume

Event Volumes have been introduced in [9]. In the original work, the contribution from the two polarities is subtracted, here instead we consider each polarity independently. Given the input events

Figure 2: Visualization of a timesurface with a constant decay of 100 ms. Large values are represented with warm colors, small values with cool colors.

$\{e_i = (x_i, y_i, p_i, t_i)\}_{i=1}^{I}$, the corresponding Event Volume $V$ is given by:

$$V_{t,p,x,y} = \sum_{i, x_i=x, y_i=y, p_i=p} \max(0, 1 - |t - t_i^*|) \tag{6}$$

$$t_i^* = (B - 1)\frac{t_i - t_0}{t_I - t_1} \tag{7}$$

We omit the bilinear kernel for x, y as we restrict to build the volume at maximum resolution. We simply downsample after with a dense operator. In the experiments, we generate Event Volumes of $B = 5$ bins and 2 polarities (for ON and OFF events). The input tensor shape used in the experiment is $(10, M, N)$, where bins and polarity are combined in the first dimension.

## 5 Neural Network Architecture

The proposed neural network **RED** consists of a feature extractor that is fed to bounding box regression heads. The feedforward part of the proposed feature extractor uses Squeeze-Excite Layers, denoted SE and described in Tab. 3. The ConvLSTM uses a BatchNorm with Conv Layer for input-to-hidden connection, and a plain Conv Layer for hidden-to-hidden connection. We run input-to-hidden connections (including the BatchNorm layer) in parallel for all timesteps of a batch. Detailed number of layers and parameters for each layer are given in Tab. 2.

Table 2: Architecture of the Feature Extractor.

|  | Layer Type | Kernel size | Channels Out | Stride |
|---|---|---|---|---|
| Layer1 | BNConvReLU | 7 | 32 | 2 |
| Layer2 | SE | 3 | 64 | 2 |
| Layer3 | SE | 3 | 64 | 2 |
| Layer4 | SE | 3 | 128 | 2 |
| Layer5 | ConvLSTM | 3 | 256 | 2 |
| Layer6 | ConvLSTM | 3 | 256 | 2 |
| Layer7 | ConvLSTM | 3 | 256 | 2 |
| Layer8 | ConvLSTM | 3 | 256 | 2 |
| Layer9 | ConvLSTM | 3 | 256 | 2 |

Table 3: Architecture of the Squeeze-Excitation (SE) Layers.

|  | Layer Type | Kernel size | Channels Out | Stride |
|---|---|---|---|---|
| Layer1 | BNConvReLU | 3 | 256 | 2 |
| Layer2 | BNConvReLU | 3 | 256 | 1 |
| Layer3 | BNConv | 3 | 256 | 2 |
| Layer4 | GlobalAvgeragePooling | MxN | 256 | MxN |
| Layer5 | DenseReLU | 1 | 64 | 1 |
| Layer6 | DenseSigmoid | 1 | 256 | 1 |
| Layer7 | Elementwise-Multiplication | 1 | 256 | 1 |
| Layer8 | Skip-Sum | 1 | 256 | 1 |