[Reviews · NeurIPS 2020]

Review 1

Summary and Contributions: In this paper, the authors first introduce a new large-scale benchmark for event-camera object detection. The main characteristics include 1) large-scale 2) high-resolution. The extensive experiments create a solid baseline for event-based object detection. Then, a novel architecture for this task is proposed. The main idea is the mix of low-level convolutional features, following by ConvLSTM layers for high-level temporal aggregation. Meanwhile, a temporal consistency loss is leveraged to enhance the temporal consistency of the detection cross frames.

Strengths: 1) novel dataset: large-scale, high-resolution The proposed dataset fill in the blank of event-based object detection benchmark. The lack of a large-scale high-resolution dataset hinders the way of the development of the related detection architecture. 2) new architecture: intuitive usage of existing building blocks The authors follow some popular practices in 2D object detection in RGB images, as well as the ConvLSTM design for spatial-temporal feature extraction. 3) extensive experiments: solid baseline, insightful ablation study As for experimental section, the authors provide enough details and discussions to provide not only the big picture, but also practical comparison to the readers. Such results could help the community to rethink the importance and potential of event-based object detection.

Weaknesses: 1) the lack of substantial technical novelty Even though the authors propose both dataset and method in this submission, the lack of the fundamental technical contribution is still a drawback of this paper. The authors borrow the convolutional architecture from 2D object detection, temporal loss/module from event analysis community. 2) More discussion on labeling protocol It could be much better if the authors could provide more details on labeling procedure. Currently, the authors mainly describe how to transfer the annotation from RGB to event camera. At the same time, labeling on RGB images with temporal/motion clues could be an interesting topic itself.

Correctness: The experimental setup looks good to me. The main metric follows the practice of COCO detection benchmark. The ablation study provides several interesting and necessary groups of experiments to show the design choices and input representations.

Clarity: The overall writing looks good to me. The main storyline looks consistent. The crucial details are included in the main submission.

Relation to Prior Work: The authors discuss the related works of event-based object detection architecture in a detailed way. As for dataset perspective, the authors include several relevant datasets and compare the proposed dataset with the existing ones.

Reproducibility: Yes

Additional Feedback:


Review 2

Summary and Contributions: This submission proposed an end-to-end learning-based method to detect moving objects with an event camera. The method is evaluated on a new dataset consisting of events and videos with a labeled bounding box, with a number of useful ablations and comparisons.

Strengths: This manuscript is well written and technically sounds. A temporal consistency loss and recurrent architecture have been introduced in this submission. A high-resolution and large scale dataset has been collected for training by the author.

Weaknesses: Failure cases are missing. As the event camera only records intensity changes, the proposed method is mainly targeting moving objects. It will be better to highlight `moving objects' in the title or somewhere else.

Correctness: Yes, the claims and method correct.

Clarity: Yes.

Relation to Prior Work: Yes.

Reproducibility: Yes

Additional Feedback: 1. We would like to see some discussions on failure cases. For example, image-based methods might fail in extreme lighting conditions. Will the proposed method works stable if the camera system mounted with a slowly moving car? 2. We would like to know more details about Events-RetinaNet. Based on the current version, it seems like use RetinaNet on the integrated event frame. If so, we would like to know how the event frame (Fig 4, top row) is estimated. If not, we would like to know the result of the integrated event frame (1 or 2 ways) + RetinaNet. 3. Fig. 2, h_{0, k+1} or h_{0, k-1} at the bottom?


Review 3

Summary and Contributions: This paper acquired and released a high-resolution and large-scale dataset for event-based object detection. The authors designed an event+RGB camera setup and proposed a fully-automated method to label event data from RGB frames. It proposed a network architecture for event-based object detection, which used the temporal consistency loss to obtain more precise localization overtime. The qualitative experiments show that this detector can achieve better performance than SOTA frame-based object detection methods.

Strengths: + The high-resolution event-based detection dataset is meaningful for the future research; it provides an effective dataset for future research on deep learning-based methods. The fully automated labeling protocol, the time synchronization and spatial registration methods are very interesting attempts. + The proposed recurrent network can detect objects directly from event data, with the same accuracy as SOTA frame-based detectors. Processing event data directly helps to reduce computational consumption, reduce the amount of parameters, and improve detection efficiency. + The authors did thorough ablation experiments that compared the effectiveness for different event representations and losses. The authors also did sufficient comparison experiments, they used COCO metrics to compare the detection accuracy between frame-based and event-based detectors, the results show that the RED can detect objects more accurately in a shorter time. The authors also verified the generalization capabilities of their detector on night recordings (HDR).

Weaknesses: - It is suggested that the authors should add a comparison of the amount of data into Table II to show the advantages of RED compared with Gray-RetinaNet in terms of data storage efficiency. - The author can process data from DAVIS and other event cameras to prove the compatibility of the proposed detector with current mainstream event cameras. - It is better to discuss limitations of the proposed method in extreme scenarios, such as the scenario with a large number of overlapping objects.

Correctness: Yes, the claims and method are correct, and the authors proved the correctness by comparative experiments.

Clarity: I enjoyed reading this paper. It's a well-written work.

Relation to Prior Work: Yes, this paper clearly discussed the difference between the proposed RED and two existed event-based object detection works.

Reproducibility: Yes

Additional Feedback: See my comments about weakness. It seems that such a 1M pixel event camera is not available on market yet. I hope the data could be released upon acceptance to inspire future research. ------------------------------ I am satisfying the with rebuttal and keep my positive rating unchanged.


Review 4

Summary and Contributions: The work presents a solid approach to object detection (in particular for street scenes) for 1 mega-pixel event cameras. This type of camera does not capture a full image frame at a certain frequency, but only creates asynchronous events for those pixels that change in intensity. The proposed detection architecture combines established components to compute features, keep state over time (something that is a hard requirement for event cameras) and drive detection. The training is done on a large, novel and to-be-published dataset, which has been collected in a scalable way. Furthermore, the approach is thoroughly compared against frame-based approaches and strong baselines and exhibits a better performance.

Strengths: - Object detection, especially for traffic participants in street scenes is an important problem on its own. Using an event camera enables faster -- possibly more efficient -- vision with a higher dynamic range. A successful approach would enable for example autonomous driving in dim lighting conditions potentially at a lower energetic cost. - The methodology to collect the dataset is insightful: The authors build a two-camera rig combining a frame-based with an event-camera and use a label transfer mechanism to label 15M bounding boxes for ~15h of recorded video of diverse scenarios. - The authors plan to release their collected dataset. This would further support the development of detection algorithms for event cameras and to measure progress. It is the largest dataset that I have seen so far. - The object detector is well tailored towards the characteristics of event cameras. In particular, great care went into event representation, evaluation of detection and smoothing detections over time. - The proposed approach is pretty unique (due to lack of previous training data) and there are not too many other works to compare against. However, the authors go to great lengths to build competitive baselines to show-case their method and compare against frame based methods as well. - The method out-performs the presented baselines by a large margin and is on-par with a frame-based approach. However, the event camera based approach shines in low-light conditions and exhibits a finer time-resolution.

Weaknesses: - I believe that the details given in this work are detailed enough to reproduce the experimental results in the paper. The used neural network layers/functions are well established and the training schedule looks clean. However, I would like to encourage the authors to publish their code. - This work might not be a paradigm shift in methodology. However all components are well chosen from previous published work. Examples include the representation of events based on [48, 22, 49, 23], keeping temporal state from [39] or the detector head from [37]. - Resolution is still limited by the sensors of event camera and there is no color information (Although this sensor resolution is certainly interesting for driving video streams). During the experimental comparison, frame-based methods are also down-scaled to the same resolution as the event camera and converted to grayscale. It is exciting to see that the performance for event cameras is on-par with frame-based cameras (in bright settings!), however a camera sensor with a higher resolution and color is likely performing better. This is not necessarily a weakness for the method; it just leads to potential missing empirical evidence that event cameras in the proposed setup would beat frame-based methods in general. - I might have missed this, but the work does not talk about verification that the automated label transfer of object bounding boxes indeed does work to a certain degree. The qualitative examples (and the detection rates) seem to be evidence of the high quality. However, I would have liked some discussion about this issue.

Correctness: Yes.

Clarity: The paper is well written and easy to follow. The authors make sure that uncommon aspects are well introduced. Experimental results are discussed clearly and fairly.

Relation to Prior Work: Yes, I believe so.

Reproducibility: Yes

Additional Feedback: Additional questions to the authors: - Do you plan to publish your implementation of the experiments along with the dataset? - Do you plan to publish evaluation code along with the dataset? This would ensure that future research is evaluated in the same way as your method. - Have you considered using an event queue per pixel of the past events instead of the mentioned histogram, time surface or event volumes? The work of "Learning an event sequence embedding for dense event-based deep stereo" by Tulyakov et al. report success representing past state by collecting events per pixel rather than training some RNN-like architecture to keep state over time. This might be in line with your final thoughts in the conclusion. - What is the performance of "Gray-RetinaNet" without the resolution and color alternations? - How did you validate the automatically transferred labels in the dataset? (low priority: - What do you think hinders computer vision on event cameras to surpass frame-base camera reasoning?) -------- I would like to thank the authors for their rebuttal and the insightful reviews. I recommend to publish this work. Even more, as the authors promise to make the evaluation protocol clear and append the comparison with a non-downscaled and colored result for the frame based method.

[Author Response · NeurIPS 2020]

We would like to thank the reviewers for their comments. In the following, we address the points that were raised.

**Analysis of Failure Cases (R2/R3).** *(R2) Failure cases are missing. As the event camera only records intensity*
*changes, the proposed method is mainly targeting moving objects. It will be better to highlight 'moving objects' in*
*the title or somewhere else. (R3) It is better to discuss the limitations of the proposed method in extreme scenarios,*
*such as the scenario with a large number of overlapping objects.* We agree with the reviewers that a more detailed
discussion of failure cases will add insights into the robustness of the proposed approach. We will be happy to add such
a section to the revised version of our submission. However, we would like to stress that our method is not limited to
moving objects. As shown in Fig. 4, the memory mechanism introduced by the recurrent layers of our model, is able to
consistently detect objects even when their motion does not generate events anymore.

**Technical Novelty (R1/R4).** *(R1) The authors borrow the convolutional architecture from 2D object detection,*
*temporal loss/module from event analysis community.* Although the core blocks of our model (ConvLSTM, SSD
head) are well-known layers from the frame-based community, the architecture we propose is original and designed
to work specifically for event data. Our design choices make our method efficient enough to process high resolution
event cameras and accurate enough to reach comparable accuracy as frame-based detectors. Moreover, even if we find
inspiration in works such as [43, 44], to the best of our knowledge, our work is the first to introduce a double regression
head and a temporal consistency loss, well adapted to the characteristics of event data. Finally, as also pointed by R3,
we believe that the automated labeling protocol is an additional technical contribution that allows fast creation of large
event-based datasets. We hope this will further extend the impact of our work for the community.

**Details on Labeling Protocol (R1/R4).** Due to space limitations, many details about the labeling protocol have been
described only in the supplementary material. We will be happy to add them to the main body of the revised article. In
particular, we will discuss in more depth quality assessment and failure cases.

**Dataset and Code Release (R3/R4).** We confirm that the dataset will be released upon acceptance of the paper.
Moreover, we will be glad to also release the evaluation code together with the dataset. However, to date, we can not
commit on the release of the training code, since this requires further internal discussions.

**Additional comments.** *(R2) more details about Events-RetinaNet. Based on the current version, it seems like*
*use RetinaNet on the integrated event frame.* Events-RetinaNet uses Event Volumes as input, which are the input
representation leading to the best results in our experiments and are also the same input used by our RED architecture.

*(R2) Fig. 2, $h_{0,k+1}$ or $h_{0,k-1}$ at the bottom?* Yes, there is indeed a typo in the figure caption. We thank the reviewer for
pointing it out.

*(R3) comparison of the amount of data [...] in terms of data storage efficiency.* We report in Tab. 2 the amount of
networks parameters. This shows that our network is smaller than Gray-RetinaNet. In terms of input data storage, our
method does not take advantage of the input data sparsity yet. We briefly discuss this in Sec. 6. We will extend this
analysis in the revised version of our submission.

*(R3) process data from DAVIS and other event cameras to prove the compatibility.* Indeed, our approach is not limited
to a particular type of event camera and can be applied to any event sensor. However, since there is no available DAVIS
dataset with object detection labels, we could not train our model on a DAVIS camera. Nevertheless, following the
reviewer remark, we run some qualitative tests by applying a model trained on the ATIS dataset (which has similar
resolution as the DAVIS) on the MVSEC dataset (DAVIS-346) [32]. Even if the model was trained on ATIS data, we
observe that it generalizes well also on the DAVIS dataset. We will be happy to add these results in a revised version of
the paper.

*(R4) Have you considered using an event queue per pixel (Tulyakov et al.) [...] rather than training some RNN-like*
*architecture.* Our method is agnostic to the input representation and could be indeed combined with Tulyakov et al. We
will add this remark and this reference to our submission. However, in our work, we decided to introduce the memory
mechanism in the deeper layers of the network, rather than at the pixel level. The reason for this is that it would be too
expensive to have per pixel memory states, especially for a 1Mpx event camera. Moreover, in this way, we can learn
memory states corresponding to high-level object features, which vary slowly and allow smoother detections over time.

*(R4) "Gray-RetinaNet" without the resolution and color alterations.* In our experiments, we used gray-level images
at the same resolution as the event camera. This was done to remove as much as possible a performance bias due to
information which is not available to the event camera. However, we agree that a comparison with a typical frame-based
camera is also interesting on its own. From some preliminary tests, it seems that color increase mAP by a few percentage
points. We will add and clarify this aspect in a revised version of the paper.

[Meta-Review · NeurIPS 2020]

This paper introduces dataset and benchmark for object detection in high-resolution event cameras. Reviews praise novelty of the problem, the dataset, the baseline architecture, and thoroughness of experimental results. Weaknesses are the that the baseline is mostly composed of off-the-shelf parts and some minor points which were mostly addressed in the rebuttal. The final reviews are unanimously positive, so this looks like a clear accept.